# Co-Occurrence of Mycotoxins in the Diet and in the Milk of Dairy Cows from the Southeast Region of Brazil

**DOI:** 10.3390/toxins16110492

**Published:** 2024-11-15

**Authors:** Aline Moreira Borowsky, Roice Eliana Rosim, Fernando Gustavo Tonin, Carlos Augusto Fernandes de Oliveira, Carlos Humberto Corassin

**Affiliations:** 1Departament of Animal Science, Luiz de Queiroz College of Agriculture (ESALQ), University of São Paulo, Av. Pádua Dias, 11, Piracicaba 13418-900, SP, Brazil; alineborowsky@usp.br; 2Department of Food Engineering, Faculty of Animal Science and Food Engineering (FZEA), University of São Paulo (USP), Av. Duque de Caxias Norte, 225, Pirassununga 13635-900, SP, Brazil; roice@usp.br (R.E.R.); fgtonin@usp.br (F.G.T.)

**Keywords:** mycotoxins, occurrence, total mixed ration, milk, residues

## Abstract

Mycotoxins are toxic fungi secondary metabolites that develop on feedstuffs and can be transferred into milk, thus representing a public health risk. The objective of this study was to assess the co-occurrence of mycotoxins in the diet and in the milk of dairy cows from the southeast region of Brazil. Samples of total mixed ration (TMR, *n* = 70) and milk (*n* = 70) were collected in dairy farms and subjected to multi-mycotoxin analysis using liquid chromatography coupled to tandem mass spectrometry. The aflatoxins (AFs), ochratoxin A (OTA), and T-2 and HT-2 toxins were not detected in TMR samples. In contrast, fumonisins (FBs), zearalenone (ZEN), and deoxynivalenol (DON) were detected in 100, 93, and 24% of TMR samples at mean levels of 336.7 ± 36.98, 80.32 ± 16.06 µg/kg and 292.1 ± 85.68 µg/kg, respectively. Ninety-two percent of TMR samples exhibited co-occurring mycotoxins. In milk, 54% of samples (*n* = 38) had detectable levels of mycotoxin, while 43% (*n* = 30) contained two or more types of mycotoxins. DON, FB, and ZEN metabolites (α-zearalenol and β-zearalenol) were the most frequent mycotoxins detected in the milk samples analyzed, at mean concentrations of 0.562 ± 0.112, 2.135 ± 0.296 µg/kg, 2.472 ± 0.436 µg/kg, and 0.343 ± 0.062 µg/kg, respectively. However, none of the analyzed milk samples had levels higher than the maximum permitted limit for AFM_1_ in Brazil (0.5 µg/L). The results of this trial highlight the concern about the co-occurrence of multiple mycotoxins in TMR and in milk, due to the possible additive or synergistic effects of these compounds. The presence of co-occurring mycotoxins in milk underscores the need for stringent preventive practices to avoid mycotoxin contamination in the diet of dairy cows in Brazil.

## 1. Introduction

Mycotoxins are toxic compounds produced by the secondary metabolism of fungi and have been associated with several adverse effects on human and animal health, as documented in various studies [1]. These compounds have been linked to carcinogenic, teratogenic, and immunotoxic effects, among others. In animals, the effects of mycotoxins have been observed to include reduced performance, reduced milk production, and weight loss [2].

Animals become contaminated by ingesting feed that has been contaminated with mycotoxins in their diet. Although ruminants are more resistant due to the presence of the rumen, which acts as a protective barrier against mycotoxins, dairy cows are more at risk from dietary contamination [3]. The total mixed ration (TMR) is the predominant feeding method for high-production dairy cows. TMR contains varied proportions of silage (the main dietary component), raw materials, and co-products such as cottonseed, cereals, protein supplements, vitamins, and minerals [4]. Among these ingredients, silage and cereals are the main products susceptible to fungal development and mycotoxin production, especially during improper storage conditions [4,5]. Moreover, the heterogeneous composition of TMR ingredients leads to variations in the frequencies and levels of mycotoxins in this product, also favoring the simultaneous occurrence of different mycotoxins in a single batch of TMR. For example, of the 74,821 feed and ingredient samples collected in multiple regions of the world between 2008 and 2017, 88% were contaminated with at least one mycotoxin, while 64% contained at least two mycotoxins [6]. The main mycotoxins identified in TMR include regulated compounds such as fumonisins (FBs), zearalenone (ZEN), and deoxynivalenol (DON) [7]. Additionally, non-regulated mycotoxins such as nivalenol, enniatins, and beauvericin have been frequently reported in TMR, although they remain largely underestimated in dairy farming practices [8]. The data presented serve to highlight concerns regarding the co-occurrence of mycotoxins, a phenomenon that is becoming increasingly prevalent.

In Brazil, the tropical characteristics provide suitable climate conditions for the growth of various species of mycotoxin-producing fungi [9]. The combination of high temperatures and elevated humidity creates an environment that facilitates the proliferation of these organisms, thereby increasing the risk of contamination of the feed with mycotoxins. Studies have shown the frequent occurrence of aflatoxins (AFs) and ochratoxin A (OTA) in brewer’s grains, finished cow’s feed, corn, and barley rootlets intended for dairy cows [10,11]. In TMR samples, the presence of various mycotoxins has also been documented, including AFs, FBs, DON, OTA, ZEN, and T-2 toxin [12].

Mycotoxins can be transferred to animal-derived products [3]. Consequently, mycotoxin-contaminated milk represents a public health risk as it has the potential to serve as an additional source of mycotoxin exposure to humans. In milk, the focus has traditionally been on the presence of aflatoxin M_1_ (AFM_1_), the hydroxylated metabolite originating from the biotransformation of aflatoxin B_1_ in the liver of lactating animals [2]. AFM_1_ is a potentially carcinogenic compound that has been studied extensively in milk and its derivatives [13,14,15]. However, in recent years, other mycotoxins have been found in milk, such as deoxynivalenol (DON), ochratoxin A (OTA), and zearalenone (ZEN) [16]. The number of studies in this area is still limited, but it is becoming increasingly necessary to consider the potential for synergistic effects and the potential of fungi to produce multiple types of mycotoxins. In addition, most mycotoxins in milk are not regulated, and, when they are, co-occurrence is often not considered. In this scenario, the present study aimed to identify and quantify the mycotoxins present in the diet and milk of dairy cows from the southeast region of Brazil, to verify the possible transfer of these metabolites into milk.

## 2. Results

### 2.1. Characterization of Total Mixed Ration

The TMRs subjected to analysis had a wide range of ingredient compositions, with a majority (34.3%) containing five or more ingredients. The most common ingredients in the samples were identified as corn silage, soybean meal, soybean hulls, corn, and cottonseed, as shown in Table 1. The ingredients were grouped according to the following categorizations to facilitate the classification of the components present in the TMR: cottonseed, soybean hulls, soybean meal, hay, cornmeal, dried distillers grains (DDGs), wet corn and soybeans, corn (ground and rehydrated), citrus pulp, pre-dried (tifton, ryegrass, alfalfa, oat grass, and oats), commercial feed, corn silage, wet corn grain silage, soybeans, other concentrates (extruded soybean meal, potatoes, bulgur, oats, wheat straw), other meals (wheat, cotton, and corn gluten), other silages (wheat, sorghum, triticale, and grass), and other roughages (sorghum, barley, alfalfa, and malt rootlets).

### 2.2. Mycotoxins in Total Mixed Ration

The concentrations of mycotoxins in TMR are presented in Table 2. AFs, OTA, and T-2 and HT-2 toxins were not detected in the TMR samples. However, FBs (sum of FB_1_ and B_2_), ZEN, and DON were detected at mean levels ranging from 80.32 to 336.7 µg/kg, with a considerable range of variation, ranging from 2.25 to 2372 µg/kg. All positive samples had concentrations below the European Union (EU) thresholds for FBs and DON.

In the present study, no significant correlation was found between the occurrence of mycotoxins in the TMR samples (*p* > 0.05), as indicated in Table 3. However, all TMR ingredients evaluated had a 100% occurrence of FBs (Table 4).

Regarding the co-occurrence of mycotoxins in TMR, 67 and 25% of samples analyzed contained two (FB and ZEN) or three (FBs, ZEN, and DON) mycotoxins, respectively, as shown in Figure 1.

### 2.3. Mycotoxins in Milk

The concentrations of mycotoxin residues in milk samples are presented in Table 5. The presence of AFM_1_ was observed in only 3% of the milk samples, at levels ranging from 0.03 to 0.320 µg/L. Furthermore, 50% of the samples analyzed showed the presence of FBs (sum of FB_1_ and FB_2_), α-zearalenol (α-ZEL), β-zearalenol (β-ZEL), or DON, at mean levels of 2.135 ± 0.296, 2.472 ± 0.436, 0.343 ± 0.062, and 0.562 ± 0.112 μg/L, respectively.

Importantly, 43% (*n* = 30) of the milk samples evaluated had more than one type of mycotoxin, with the majority containing two and three mycotoxins, as presented in Table 6. The most common co-occurrence was that of α-ZEL + β-ZEL + FBs, which constituted 10% of the total occurrences, followed by a combination of α-ZEL + β-ZEL + DON (9%). The correlations between mycotoxins in the TMR and milk samples evaluated in this study are presented in Table 7.

## 3. Discussion

### 3.1. Mycotoxins in Total Mixed Ration

The contamination of TMR with mycotoxins is a consequence of the presence of mycotoxin-producing fungi during the cultivation, storage, and processing of feed ingredients. Several fungal species have the capacity to produce mycotoxins at different stages of the production steps of cultivated crops, including the growth, post-harvest, storage, transportation, and processing phases [17]. Environmental factors, including temperature, humidity, and insect activity, have been identified as significant contributors to the proliferation of fungi and the subsequent mycotoxin production in cereals [18], which are important ingredients of TMR. In the present study, AFs were not detected in any sample of TMR, although two milk samples contained AFM_1_ at levels of 0.03 and 0.320 µg/L. Most dairy farms evaluated (91%) used anti-mycotoxin additives such as mineral adsorbents, which are often only effective against the AFs, and this may at least partly explain the lack of detection of this mycotoxin in the TMR samples. However, studies conducted worldwide have found AFs in up to 100% of TMR samples intended for dairy cows [4,5,19,20,21,22,23,24]. In contrast, the detection of OTA, and HT-2, and T2 toxins is less frequent in studies, with occurrence records as high as 34.7% for OTA and HT-2 in Costa Rica [25] and 100% for T2 in South Africa [20]. FBs and ZEN are frequently found in a significant proportion of TMR samples, while DON tends to be comparatively less common, according to other studies [26,27,28].

As reported by Martins et al. [8], the most frequently identified mycotoxins in TMR are those that are subject to regulation: aflatoxins, ZEN, DON, T2, and FB. However, it is recognized that geoclimatic factors, such as temperature and relative humidity, as well as the specific composition of TMR, may contribute to the observed variations [5,29]. In Brazil, Custódio et al. [30] investigated TMR consumed by beef cattle in the southeast and midwest regions, and observed that the prevalence of FB mycotoxins was the highest (93.3%). Type A trichothecenes, including T2 and HT-2 toxins, were found in 66.7% of the samples, while aflatoxins and type B trichothecenes such as DON were present in 6.7% and 20.3%, respectively. In addition, ZEN and OTA were not detected, and only 3.3% of the properties used additives, indicating that climate, diet composition, and additive use influence the results obtained. Biscoto et al. [12] reported that the predominant mycotoxins in Brazilian TMR samples were ZEN (77.5%), DON (70.3%), and AFs (65.7%); The maximum observed limits for these mycotoxins were 1450 µg/kg, 4829 µg/kg, and 266.6 µg/kg, respectively, which were higher than those found in the present study.

It is important to note that all positive samples had concentrations below the thresholds established by the EU for FB and DON, which are 20,000 µg/kg and 2000 µg/kg, respectively [31]. Nevertheless, three percent of the samples showed values above the EU limit for zearalenone, which is set at 500 µg/kg, with concentrations ranging from 596.15 to 615.35 µg/kg. All 70 samples that were subjected to the evaluation contained at least one mycotoxin, with FBs being a particularly prevalent contaminant. Moreover, it was observed that 67% of the samples contained the simultaneous presence of two mycotoxins (FB and ZEN), and 25% had three mycotoxins (FB, ZEN, and DON), as displayed in Figure 1. The high percentage of samples (92%) with co-occurring mycotoxins reflects the growing public health concern about contamination by these compounds. In Mexico, an average of 24 mycotoxins per TMR sample was reported, with a range of 9 to 31 mycotoxins per sample [26]. Meanwhile, in Thailand, between 20 and 44 metabolites were identified in TMR samples, with the most common combination observed being between ZEN and FB1, followed by the combination of ZEN and DON [27]. Several studies have addressed the interaction effects between mycotoxins such as AFs and OTA or FBs in farm animals such as piglets and sheep [16]. However, thus far, the consequences of dairy cows’ exposure to dietary co-occurring mycotoxins have not been established.

In agreement with the data presented here, a total of 64% of the 74,821 samples of feed and feed ingredients collected from 100 countries between 2008 and 2017 were found to be contaminated with two or more mycotoxins [6]. This prevalence is of concern, especially considering the potential additive effects of these mycotoxins, which could increase their negative impact on animals. Muñoz-Solano and González-Peñas [32] investigated the co-occurrence of mycotoxins in diets for different animal species and found weak correlations (*p* ≥ 0.05) between ZEN and aflatoxin. For AFB_2_, G_1_, and G_2_, as well as between DON and AFG_1_, a weak correlation was observed. In addition, a moderate correlation was observed between AFB_1_ and AFB_2_, AFG_1_ and AFG_2_, and DON and ZEN. However, no significant correlation was found in the present study [32].

In a study conducted in Kenya on the presence of mycotoxins in dairy cow diets, a high occurrence of FB_1_ and FB_2_ was observed, with frequencies of 100 and 94%, respectively [33]. In Mexico, no correlation was found between the presence of FB, ZEN, and type B trichothecenes with the ingredients used [26]. However, the researchers observed a positive correlation between the presence of concentrates and ergot alkaloids, corn straw, and the *Fusarium* genus, as well as the overall presence of mycotoxins. Conversely, sorghum silage showed a correlation with FB. In our study, ZEN contamination rates in various feed products ranged from 66.7% in wet corn silage and other concentrates to 100% in corn meal, wet corn, and soybeans, pre-dried, commercial feed, soybeans, and other meals. It is important to note that ZEN is predominantly found in cereals, including corn, wheat, rice, barley, sorghum, soybeans, oats, and their derivatives [34]. In contrast, DON was not observed in wet corn silage and other roughages but was present in 50% of the samples of other concentrates. DON is a mycotoxin found in pastures and silages, as well as in cereals such as wheat, corn, barley, rye, and oats [19].

### 3.2. Mycotoxins in Milk

It is well-established that mycotoxins ingested by animals can undergo biotransformation or remain unmetabolized, with both forms being potentially excreted into milk. The biotransformation process can result in the formation of less toxic metabolites that are subsequently excreted in milk [3]. However, the presence of unmetabolized mycotoxins in milk poses a significant risk to human health. The consumption of mycotoxin-contaminated milk can lead to indirect exposure to these toxic compounds, thus increasing the risk of adverse health effects, particularly in vulnerable populations such as children. Such effects may include impaired growth and development, toxicities affecting various tissues—including the immune system—and potentially severe outcomes such as cancer or even death [13]. Additionally, it is recognized that mycotoxins can alter nutritional factors in milk, leading to a decrease in both the production and quantity of fat and protein content [35]. Among the main mycotoxins excreted in milk, AFM_1_, a compound resulting from the hepatic biotransformation of AFB_1_, stands out [36]. It is estimated that approximately 5% of ingested AFB_1_ is converted to AFM_1_ and excreted in milk [37]. However, the amount of AFM_1_ excreted is subject to individual variations and seasonal variation, depending on the composition of the diet in both summer and winter [38]. Several studies conducted in different regions of the world have documented the presence of AFM_1_ in milk samples from different species. For instance, Abdallah et al. reported that all cow’s milk samples from Egypt were contaminated with AFM_1_, with concentrations ranging from 0.02 to 0.19 μg/kg [38]. Furthermore, 70% of these samples exceeded the European Union limit of 0.05 μg/L [39]. Similarly, in Costa Rica [26], 53.5% of cow’s milk samples were contaminated with AFM_1_ between 2014 and 2017, with concentrations ranging from 0.005 to 0.989 μg/L. Only 1.3% of the samples exceeded the 0.5 μg/L limit established by US legislation [40]. Globally, approximately 10% of milk samples analyzed exhibit levels of AFM_1_ exceeding the tolerance limit established by the European Union [41].

In the current study, the presence of AFM_1_ was observed in only 15% of the milk samples, which may indicate possible limitations of the sampling procedures used, especially considering that aflatoxins were not detected in the feed. The presence of other mycotoxins in milk (FB, α-ZEL, β-ZEL, and DON) is consistent with the trends observed in recent years, indicating the occurrence of other mycotoxins in milk, in addition to AFM_1_. For example, in Nigeria, aflatoxin P_1_, alternariol monomethyl ether, citrinin, dihydrocitrinone, enniatins, ochratoxin α, and sterigmatocystin were detected for the first time in milk from both goats and cows [14]. In Portugal, although no aflatoxin types were detected in the milk samples, beauvericin (BEA) was present in 100% of the samples, enniatin B1 in 75%, FB_2_ in 30%, and FB_1_ in 10% [42].

The susceptibility of animal species to aflatoxin contamination and transfer may influence the levels of AFM_1_ found in milk. In Iraq, AFM_1_ levels were found to be significantly higher in goat and sheep milk than in cow milk and even higher than in buffalo milk. Although AFB_1_ was found at higher levels in buffalo feed than in cows, sheep, and goats, it is suggested that the transfer rate to the milk of sheep and goat milk is higher than to cow and buffalo milk [15]. In Nigeria, AFM_1_ was detected in 100% of cow milk samples and in 49% of goat milk samples; Of the goat milk samples, 13% exceeded the European Union limit, while 55% of the cow milk samples exceeded this limit; AFM_1_ was not detected in camel milk [14]. In the present study, only one sample of milk had an AFM_1_ concentration above the European Union limit of 0.05 µg/L [39]. In Yemen, 50% of the milk samples exceeded the EU limit [43], while, in Serbia, 56.3% of the samples exceeded the regulatory limit [44]. These findings underscore the importance of mycotoxin research and regulations.

The available studies on the carry-over of mycotoxins from the diet of dairy cows into milk are scarce. In Portugal, 13 different mycotoxins have been identified in raw milk; the presence of AFB_1_, AFB_2_, AFM_1_, BEA, enniatins (enniatin A and B), FB_1_, FB_2_, HT-2 toxin, moniliformin, patulin, T-2 toxin, and ZEN was confirmed in the samples, with positivity ranging from 6 to 97% and concentrations between 0.006 and 16.32 µg/L [44]. In Brazil, Frey et al. were the first to report the co-occurrence of mycotoxins in fluid milk. The authors found de-epoxy-deoxynivalenol, OTA, FB_1_, FB_2_, α-ZEL, and β-ZEL in the samples, with mean concentrations of 0.417, 0.072, 1.613, 0.753, 0.612, and 1.334 µg/L, respectively [16]. The mean concentrations of DON, FB, α-ZEL, and β-ZEL in the present study were 0.562, 0.118, 2.472, and 0.343 µg/L, respectively. The co-occurrence of these compounds was observed in milk samples, with approximately 56% of the samples contaminated with more than one mycotoxin. In Portugal, 82% of the samples contained between 4 and 7 mycotoxins [42], while, in Brazil, 30.9% of the samples showed a co-occurrence of between 2 and 4 mycotoxins [16]. In Brazil, the most common combinations were AFM_1_ and ZEL (α-ZEL + β-ZEL) and OTA and ZEL, at 6% and 7.3%, respectively [16]. This evidence supports the hypothesis that the presence of mycotoxins is dependent on seasonal, geographical, and material-specific factors.

Some correlations between mycotoxins in the diet and milk of cows are significant. Given the limited number of studies to support the discussion of these findings, it is reasonable to conclude that the results reported here are consistent with the expectations and show a correlation between FB and DON, both present in the diet and in the milk. These findings suggest that other mycotoxins may also be transferred from the diet to the milk. Furthermore, the presence of ZEN in the diet correlated with the presence of α-ZEL + β-ZEL, confirming that these compounds are biotransformed into other metabolites by the animal’s body. The correlation between FB and DON, and between DON and β-ZEL, also highlights that fungi can produce more than one type of mycotoxin, thereby increasing the toxic effects.

The results of this study indicate a direct correlation between the level of mycotoxin contamination in animal feed and the concentration of these toxins in milk. Therefore, the higher levels of DON, FB, and ZEN contamination found in TMR corresponded to the increased transfer of residues of these mycotoxins into the milk. These findings highlight the need for stringent control measures to avoid mycotoxin contamination in both feed and dairy products in the evaluated dairy farms. The implementation of rigorous monitoring protocols and the improvement of agricultural practices are among the most important strategies for mitigating the exposure of lactating animals to dietary mycotoxins and avoiding the contamination of the produced milk with mycotoxin residues in dairy farms.

## 4. Conclusions

Although AFs were not detected in the TMR samples evaluated, the prevalence and levels of FBs, DON, and ZEN indicate a potential concern regarding the co-occurrence of these mycotoxins and the use of additives that may affect their presence. The identification of 70% of the milk samples containing AFM_1_ levels above the regulatory limit, in conjunction with the presence of other mycotoxins, highlights the concern about co-occurrence, due to the additive and synergistic effects of these compounds. The correlations between the mycotoxins identified in the diet and in the milk confirmed the transferability of certain mycotoxins through the milk, emphasizing the growing importance of preventing the presence of mycotoxins in the diet utilizing anti-mycotoxin additives capable of acting against diverse types of mycotoxins and of establishing regulations addressing the various mycotoxins present in both the diet and milk.

## 5. Materials and Methods

### 5.1. Sampling Procedures

A total of 70 samples of either TMR or milk were collected from dairy farms situated in the southeastern region of Brazil. Dairy farms were selected based on the following criteria: (a) exclusive use of TMR for animal feeding; and (b) use of a single TMR formulation for the herd, aiming at eliminating discrepancies between feed formulations across different animal groups. The number of selected farms was determined according to the percentage of milk production in the Brazilian states of southeast region, relative to the total milk produced in Brazil, as follows: Minas Gerais (75%), São Paulo (18%), Rio de Janeiro (4%), and Espirito Santo (3%) [45]. Therefore, the number of dairy farms selected in those states were 50, 15, 3, and 2, respectively.

In each dairy farm, 500 g of the available TMR were collected, transported to the laboratory, and subjected to a drying process in a forced ventilation oven at a temperature of 105 °C for 24 h, to determine the dry matter content. Subsequently, the dried TMR samples were subjected to grinding and reserved for mycotoxin analysis. Milk samples (100 mL) were also collected from the bulk tank in each dairy farm at the time of TMR collection. TMR and milk samples were stored at −80 °C until analysis.

### 5.2. Determination of Mycotoxins in Total Mixed Ration and Milk

The analyses of multiple mycotoxins in TMR (AFB_1_, AFB_2_, AFG_1_ and AFG_2_, DON, FB_1_ and FB_2_, OTA, T-2 and HT-2 toxins, and ZEN) and milk (AFM_1_, DON, FB_1_ and FB_2_, OTA, ZEN, and α-zearalenol and β-zearalenol) samples were conducted strictly following the methodologies described by [41,46], respectively, and validated previously in the laboratory [16,47].

Quantification of mycotoxins in the final extracts obtained from TMR and milk samples was conducted using a Waters Acquity I-Class ultra-performance liquid chromatography (UPLC) system (Waters Corp.) equipped with a BEH Column C18 (2.1 × 50 mm, 1.7 µm) and coupled to a Xevo TQ-S mass spectrometer (Waters Corp., Milford, MA, USA). The chromatographic conditions used were identical to those previously described for feed and milk samples by [9,39], respectively. Mass spectrometry (MS) analyses were conducted in a multiple-reaction-monitoring (MRM) mode by using electrospray ionization in a positive ion mode. MS parameters adopted and MRM transitions are displayed in Appendix A.

### 5.3. Statistical Analysis

Following the identification and quantification of the mycotoxins present in the samples, the means, standard deviations, and corresponding maximum and minimum limits were calculated using the PROC MEANS procedure of the SAS 9.4 software (SAS Institute Inc., Cary, NC, USA). Furthermore, a correlation analysis was conducted between the mycotoxins using the PROC CORR procedure. For all tests, a significance level of 5% was employed.

## Figures and Tables

**Figure 1 toxins-16-00492-f001:**
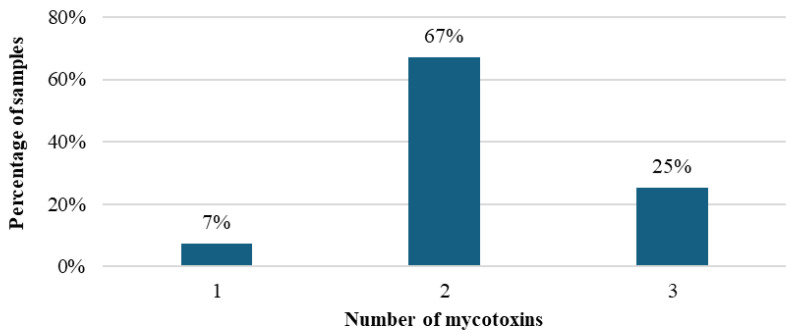
Percentage of samples of total mixed ration for dairy cows from the southeast region of Brazil containing one, two, or three mycotoxins.

**Table 1 toxins-16-00492-t001:** Ingredients determined in samples of total mixed ration (*n* = 70) consumed by dairy cows from the southeastern region of Brazil.

Ingredients	Nº of TMR Samples Containing the Ingredient	%
Cottonseed	41	61.2
Soybean hulls	49	73.1
Soybean meal	50	74.6
Hay	19	28.4
Cornmeal	5	7.5
Dried distiller’s grains (DDGs)	16	23.9
Wet corn and soy	9	13.4
Corn	47	70.1
Citrus pulp	10	14.9
Pre-dried	20	29.9
Commercial feed	6	9.0
Corn silage	66	98.5
Wet corn grain silage	3	4.5
Soybeans	14	20.9
Other concentrates	6	9.0
Other meals	5	7.5
Other silages	7	10.4
Other roughages	11	16.4

**Table 2 toxins-16-00492-t002:** Mean values, standard error (SE), and minimum and maximum concentrations of mycotoxins in total mixed ration (*n* = 70) consumed by dairy cows from the southeastern region of Brazil.

Mycotoxin	*n* (%)	Mean ± SE (µg/kg)	Minimum (µg/kg)	Maximum (µg/kg)
Deoxynivalenol	17 (24)	292.1 ± 85.68	22.42	1573
Fumonisins (B_1_ + B_2_)	70 (100)	336.7 ± 36.98	13.14	2372
Zearalenone	65 (93)	80.32 ± 16.06	2.25	615.3

*n*: number of positive samples.

**Table 3 toxins-16-00492-t003:** Pearson correlation coefficients (r) between mycotoxins detected in samples of total mixed ration (*n* = 70) consumed by dairy cows from the southeastern region of Brazil.

Mycotoxin	Deoxynivalenol	Fumonisins (B_1_ + B_2_)	Zearalenone
Deoxynivalenol	1	-	-
Fumonisins (B_1_ + B_2_)	0.227	1	-
Zearalenone	0.104	0.196	1

No significant correlations were found between values (*p* > 0.05).

**Table 4 toxins-16-00492-t004:** Mycotoxin occurrence (%) in 70 samples of total mixed ration for dairy cows in the Southeast region of Brazil by ingredients, expressed as absolute frequency percentage (number of samples).

Ingredients	Fumonisins (B_1_ + B_2_) (%)	Zearalenone (%)	Deoxynivalenol (%)
Cottonseed	100	90.2	19.5
Soybean hulls	100	91.8	28.6
Soybean meal	100	90	30
Hay	100	84.2	36.8
Cornmeal	100	100	20
Dried distiller’s grains (DDGs)	100	81.3	25
Wet corn and soy	100	100	33.3
Corn	100	91.5	27.7
Citrus Pulp	100	80	20
Pre-dried	100	100	30
Commercial feed	100	100	16.7
Corn silage	100	93.9	25.8
Wet corn grain silage	100	66.7	0
Soybeans	100	100	14.3
Other concentrates	100	66.7	50
Other meals	100	100	40
Other silages	100	85.7	28.6
Other roughages	100	81.8	0

**Table 5 toxins-16-00492-t005:** Mean values, standard error (SE), and minimum and maximum concentrations of mycotoxin residues in milk samples (*n* = 70) from dairy cows in the Brazilian southeastern region.

Mycotoxin	*n* (%)	Mean ± SE (µg/L)	Minimum (µg/L)	Maximum (µg/L)
Aflatoxin M_1_	2 (3)	0.118 ± 0.032	0.030	0.320
Deoxynivalenol	10 (14)	0.562 ± 0.112	0.020	2.960
Fumonisins (B_1_ + B_2_)	10 (14)	2.135 ± 0.296	0.460	8.240
α-zearalenol	8 (11)	2.472 ± 0.436	0.290	7.890
β-zearalenol	8 (11)	0.343 ± 0.062	0.004	1.160

*n*: number of positive samples.

**Table 6 toxins-16-00492-t006:** Mycotoxin co-occurrence in milk samples (*n* = 70) from dairy cows from the southeastern region of Brazil.

Co-Occurring Mycotoxins	Nº of Samples Positive	%
AFM_1_ + DON	4	6
α-ZEL + β-ZEL + FB	7	10
α-ZEL + β-ZEL + DON	6	9
α-ZEL + β-ZEL + AFM_1_	1	1
AFM_1_ + DON + FB	1	1
α-ZEL + β-ZEL + FB + DON	5	7
α-ZEL + β-ZEL + FB + AFM_1_	1	1
α-ZEL + β-ZEL + FB + DON + AFM_1_	5	7

**Table 7 toxins-16-00492-t007:** Pearson correlation coefficients (*r*) between mycotoxins detected in total mixed ration and in milk samples (*n* = 70 for each matrix) from dairy cows in the southeastern region of Brazil.

	Deoxynivalenol	Fumonisins (B_1_ + B_2_)	Zearalenone
Aflatoxin M_1_	−0.095	0.104	0.193
Fumonisins (B_1_ + B_2_)	−0.02	0.833 *	0.240
α-zearalenol	0.52	0.067	0.646 *
β-zearalenol	0.75 *	0.093	0.548 *
Deoxynivalenol	0.803 *	0.347 *	0.290

* Significant correlation coefficients (*p* < 0.05).

## Data Availability

The original contributions presented in the study are included in the article. Further inquiries can be directed to the corresponding author.

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
