# Peer review of "Co-Occurrence of Mycotoxins in the Diet and in the Milk of Dairy Cows from the Southeast Region of Brazil"

_toxins, 2024, doi:10.3390/toxins16110492_

Round 1

Reviewer 1 Report

Comments and Suggestions for Authors

Manuscript number: Toxins 3286160

Title: Co-occurrence of mycotoxins in the diet and in the milk of 2 dairy cows from the Southeast region of Brazil

The authors aim to assess the co-occurrence of mycotoxins in the diet and milk of dairy cows from the southeast region of Brazil. They demonstrate that the presence of co-occurring mycotoxins in milk underscores the need for stringent preventive practices to minimize mycotoxin contamination in dairy cow diets in Brazil. While this work is interesting, several concerns were found throughout the manuscript that require the authors to make major revisions. See concerns below:

Abstract:

  • Revise the abstract according to the provided review.

Introduction:

  • L 41-42: The connection between mycotoxin and the definition of TMR is not clear. Please provide one or two sentences to bridge the two.
  • L 46-49: Include additional data on this issue.
  • L 49-51: Add citations to support the claims.
  • It is important to explain the conditions in Brazil within the introduction, as the title specifies this location.

Materials and Methods:

  • L 281-283: The selection criteria for the 70 samples are unclear. Please provide information on the sampling locations, as well as farm specifications and selection criteria.

Results:

  • Clarify whether the ‘Nº of samples containing the ingredient’ refers to the number of samples or the number of samples containing the ingredient, as it is unclear.
  • In Table 4, specify the unit of measurement for Fumonisins (B1+B2), Zearalenone, and Deoxynivalenol.

Discussion:

  • The discussion primarily compares study results to other studies without explaining the mechanism by which feed becomes contaminated with mycotoxins.
  • L 171-173: Specify the table containing the high percentage of samples (93%) so readers can easily locate it.
  • Explain the biochemical impact of mycotoxins on milk nutrition.
  • Describe how and why milk can also become contaminated with mycotoxins.
  • Connect the findings on mycotoxins in feed and milk, as high levels in feed likely correspond to high contamination in milk samples. Share your interpretation.

References:

  • Verify reference formatting style and journal abbreviation consistency.

Author Response

The authors aim to assess the co-occurrence of mycotoxins in the diet and milk of dairy cows from the southeast region of Brazil. They demonstrate that the presence of co-occurring mycotoxins in milk underscores the need for stringent preventive practices to minimize mycotoxin contamination in dairy cow diets in Brazil. While this work is interesting, several concerns were found throughout the manuscript that require the authors to make major revisions. See concerns below:

Answer: Thanks for the positive comments. The manuscript was revised according to concerns raised in the review process.

Abstract:

Revise the abstract according to the provided review.

Answer: The abstract was revised to keep the numbers at minimum (please see L18-24).

Introduction:

L 41-42: The connection between mycotoxin and the definition of TMR is not clear. Please provide one or two sentences to bridge the two.

Answer: Those lines were amended in the revised manuscript, to provide a bridge between TMR and mycotoxin occurrence in this product (please see L49-54).

L 46-49: Include additional data on this issue.

Answer: Additional data was included, as requested (please see L56-60).

L 49-51: Add citations to support the claims.

Answer: Proper citations were added, as suggested (please see L58 and L60).

It is important to explain the conditions in Brazil within the introduction, as the title specifies this location.

Answer: Background information on the conditions in Brazil was amended, as requested (please see L63-70).

Materials and Methods:

L 281-283: The selection criteria for the 70 samples are unclear. Please provide information on the sampling locations, as well as farm specifications and selection criteria.

Answer: The requested information was amended in the revised manuscript, please see L333-340.

Results:

Clarify whether the ‘Nº of samples containing the ingredient’ refers to the number of samples or the number of samples containing the ingredient, as it is unclear.

Answer: It refers to the number of TMR samples containing the ingredients described in the 1st column of Table 1, as amended in the revised manuscript (please see L106).

In Table 4, specify the unit of measurement for Fumonisins (B1+B2), Zearalenone, and Deoxynivalenol.

Answer: Done.

Discussion:

The discussion primarily compares study results to other studies without explaining the mechanism by which feed becomes contaminated with mycotoxins.

Answer: The Discussion section was improved by adding the mechanisms that lead to the contamination of TMR and milk by mycotoxins (please see L159-166 and L234-245).

L 171-173: Specify the table containing the high percentage of samples (93%) so readers can easily locate it.

Answer: This percentage is not available in a table. Thus, a figure (Figure 1) was provided to indicate the percentage of samples containing co-occurring mycotoxins (please see L122-124 and L135-137).

Explain the biochemical impact of mycotoxins on milk nutrition.

Answer: This comment was addressed in the new, revised version of the manuscript (please see L243-245).

Describe how and why milk can also become contaminated with mycotoxins.

Answer: The description on how and why milk can become contaminated with mycotoxins was amended in the revised manuscript, please see L234-237.

Connect the findings on mycotoxins in feed and milk, as high levels in feed likely correspond to high contamination in milk samples. Share your interpretation.

Answer: A paragraph addressing this comment was included at the end of Discussion section (please see L309-317).

References:

Verify reference formatting style and journal abbreviation consistency.

Answer: The reference formatting style was double checked for consistency, as requested.

Reviewer 2 Report

Comments and Suggestions for Authors

The paper is a very technical study or report that quantifies the content of some mycotoxins in both diet and milk of dairy cows in a specific region of Brazil.

The toxins are quantified and detected using mass spectrometry and basically consists of a series of tables showing the type of samples and toxins obtained.

The abstract shows unnecessary numbers (even with error numbers), which makes some sentences quite unreadable. In fact, the results section also just repeat the numbers that one can see just by looking at the tables.

Discussion looks more like a review of the topic, since there is not much to discuss I suppose.

There are no mechanistic details or biochemical insights on how these toxins may affect metabolism call mom or how these toxins may have some some synergies.

English is ok. As a minor thing, line 42 should be ‘at risk from’

Author Response

The paper is a very technical study or report that quantifies the content of some mycotoxins in both diet and milk of dairy cows in a specific region of Brazil.

The toxins are quantified and detected using mass spectrometry and basically consists of a series of tables showing the type of samples and toxins obtained.

Answer: Thanks for the constructive comments.

The abstract shows unnecessary numbers (even with error numbers), which makes some sentences quite unreadable. In fact, the results section also just repeat the numbers that one can see just by looking at the tables.

Answer: The abstract and the results section were revised to keep the numbers at minimum (please see L18-24).

Discussion looks more like a review of the topic, since there is not much to discuss I suppose.

Answer: The discussion was improved, as requested (please see L159-166 and L234-245).

There are no mechanistic details or biochemical insights on how these toxins may affect metabolism call mom or how these toxins may have some some synergies.

Answer: The Discussion section was improved by adding the mechanisms on how mycotoxins affect metabolism (please see L243-245). Although possible interactions such as synergies between different mycotoxins are still a matter of scientific debate, this issue was addressed in L204-207 of the revised manuscript.

English is ok. As a minor thing, line 42 should be ‘at risk from’

Answer: Done.

Round 2

Reviewer 1 Report

Comments and Suggestions for Authors

The authors were clearly assessed, and I have no furhter comments.

Comments on the Quality of English Language

OK